# Optical tools for visualizing and controlling human GLP-1 receptor activation with high spatiotemporal resolution

Loïc Duffet[1†], Elyse T Williams[2†], Andrea Gresch[1], Simin Chen[2], Musadiq A Bhat[1], Dietmar Benke[1,3], Nina Hartrampf[2*], Tommaso Patriarchi[1,3*]

[1]Institute of Pharmacology and Toxicology, University of Zürich, Zurich, Switzerland; [2]Department of Chemistry, University of Zürich, Zürich, Switzerland; [3]Neuroscience Center Zurich, University and ETH Zürich, Zürich, Switzerland

*For correspondence:
nina.hartrampf@chem.uzh.ch
(NH);
patriarchi@pharma.uzh.ch (TP)

[†]These authors contributed equally to this work

**Abstract** The glucagon-like peptide-1 receptor (GLP1R) is a broadly expressed target of peptide hormones with essential roles in energy and glucose homeostasis, as well as of the blockbuster weight-loss drugs semaglutide and liraglutide. Despite its large clinical relevance, tools to investigate the precise activation dynamics of this receptor with high spatiotemporal resolution are limited. Here, we introduce a novel genetically encoded sensor based on the engineering of a circularly permuted green fluorescent protein into the human GLP1R, named GLPLight1. We demonstrate that fluorescence signal from GLPLight1 accurately reports the expected receptor conformational activation in response to pharmacological ligands with high sensitivity (max $\Delta F/F_0$=528%) and temporal resolution ($\tau_{ON}$ = 4.7 s). We further demonstrated that GLPLight1 shows comparable responses to glucagon-like peptide-1 (GLP-1) derivatives as observed for the native receptor. Using GLPLight1, we established an all-optical assay to characterize a novel photocaged GLP-1 derivative (photo-GLP1) and to demonstrate optical control of GLP1R activation. Thus, the new all-optical toolkit introduced here enhances our ability to study GLP1R activation with high spatiotemporal resolution.

## eLife assessment

This **valuable** Tools and Resources paper presents new tools for investigating GLP-1 signaling: a genetically-encoded sensor constructed from a mutated GLP1R receptor as well as a caged agonist peptide. The evidence for these tools working as advertised is **solid** and they may be useful for screening compounds that bind to GLP1R.

## Introduction

The glucagon-like peptide-1 receptor (GLP1R) is expressed in various parts of the brain, especially in the basolateral amygdala and hypothalamic regions (*Alvarez et al., 2005*; *Cork et al., 2015*; *Trapp and Brierley, 2022*; *Turton et al., 1996*), as well as broadly outside the central nervous system (*Campos et al., 1994*). Its endogenous ligand, glucagon-like peptide-1 (GLP-1), is a peptide, fully conserved across mammals, that carries out both central and endocrine hormonal functions for the control of energy homeostasis (*Andersen et al., 2018*). GLP-1 is produced mainly by two cell types: preproglucagon (PPG) neurons principally located in the nucleus of the solitary tract of the brain (*Trapp and Brierley, 2022*; *Turton et al., 1996*), and enterocrine cells (ECs) located in the gut (*Trapp*

and Brierley, 2022). Upon ingestion of a meal, GLP-1 is rapidly released along with gastric inhibitory polypeptide (GIP) from the gut into the bloodstream where it targets β-cells of the pancreas and stimulates the production and secretion of insulin under hyperglycemic conditions (Andersen et al., 2018). This phenomenon, known as the 'incretin effect' (Nauck and Meier, 2018), is impaired in metabolic disorders, such as type 2 diabetes mellitus (Holst et al., 2009), making GLP-1 signaling an attractive therapeutic target for the treatment of these disorders. In addition to its role in controlling satiety and food intake, central GLP-1 has also been shown to play central neuroprotective roles (Hölscher, 2022), illustrating its multifaceted role in human physiology.

The human GLP1R (hmGLP1R) is a prime target for drug screening and drug development efforts, since GLP-1 receptor agonists (GLP1RAs) have been used for decades for the treatment of type 2 diabetes and have more recently become some of the most effective and widely used weight-loss drugs (Shah and Vella, 2014). Among the techniques that can be adopted in these screening efforts are those able to monitor ligand binding to GLP1R through radioactivity-based assays (Knudsen et al., 2007) or fluorescently labeled ligands (Ast et al., 2020), and those able to monitor the coupling of GLP1R to downstream signaling pathways, for example through scintillation (Runge et al., 2003), fluorescence (Biggs et al., 2018), or bioluminescence resonance energy transfer assays (Zhang et al., 2020). A technology to directly probe ligand-induced GLP1R conformational activation with high sensitivity, molecular specificity, and spatiotemporal resolution could facilitate drug screening efforts and open important new applications (Chen et al., 2022; Frank et al., 2018), but is currently lacking.

To overcome these limitations, here we set out to engineer and characterize a new genetically encoded sensor based on the GLP1R, using an established protein engineering strategy (Duffet et al., 2022a; Patriarchi et al., 2018; Patriarchi et al., 2019; Sun et al., 2018). This sensor, which we call GLPLight1, offers a direct and real-time optical readout of GLP1R conformational activation in cells, thus opening unprecedented opportunities to investigate GLP1R physiological and pharmacological regulation in detail under a variety of conditions and systems. We demonstrated its potential for use in pharmaceutical screening assays targeting GLP1R, by confirming that GLP1R and GLPLight1 show similar ligand recognition profiles, including high specificity toward GLP-1 over other class B1 GPCR ligands, low-affinity for glucagon, and specific functional deficits of GLP-1 alanine mutants. Finally, to extend the optical toolkit further, we also developed a photocaged GLP-1 derivative (photo-GLP1) and adopted it in concert with GLPLight1 to enable all-optical control and visualization of GLP1R activation.

## Results

### Development of a genetically encoded sensor to monitor hmGLP1R activation

To develop a genetically encoded sensor based on the hmGLP1R, we initially replaced the third intracellular loop (ICL3) of hmGLP1R with a cpGFP module from the dopamine sensor dLight1.3b (Patriarchi et al., 2018), between residues K336 and T343 (Figure 1a). This initial sensor prototype had poor surface expression and a very small fluorescence response upon addition of a saturating concentration (10 μM) of GLP-1 ($\Delta F/F_0$=39%, Figure 1—figure supplement 1a). Removal of the endogenous GLP1R N-terminal secretory sequence (amino acids 1–23) from this construct improved the membrane expression and the fluorescence response to GLP-1 ($\Delta F/F_0$=107%, Figure 1—figure supplement 1a). We then performed a lysine scan on the residues spanning the intracellular loop 2 (ICL2) of the sensor. From this screening we identified one beneficial mutation (L260K) that more than doubled the dynamic range of the sensor ($\Delta F/F_0$=180%, Figure 1—figure supplement 1b). Next, we performed site-saturated mutagenesis on both receptor residues adjacent to the cpGFP and screened a subset of 95 variants. This small-scale screening led us to identification of a new variant (containing the mutations K336Y and T343N) with $\Delta F/F_0$ of about 341% (Figure 1—figure supplement 1c–d). To further enhance surface expression of the sensor, we introduced a C-terminal endoplasmic reticulum export sequence (Stockklausner and Klocker, 2003) on this variant (Figure 1—figure supplement 1e). We then introduced three previously described (Wan et al., 2021) mutations in the cpGFP moiety, which improved the basal brightness of the probe without affecting its dynamic range (Figure 1—figure supplement 1f–g). Finally, we mutated eight phosphorylation sites on the C-terminal domain that are responsible for GLP1R internalization (Thompson and Kanamarlapudi, 2015) (S431A, S432A, T440A,

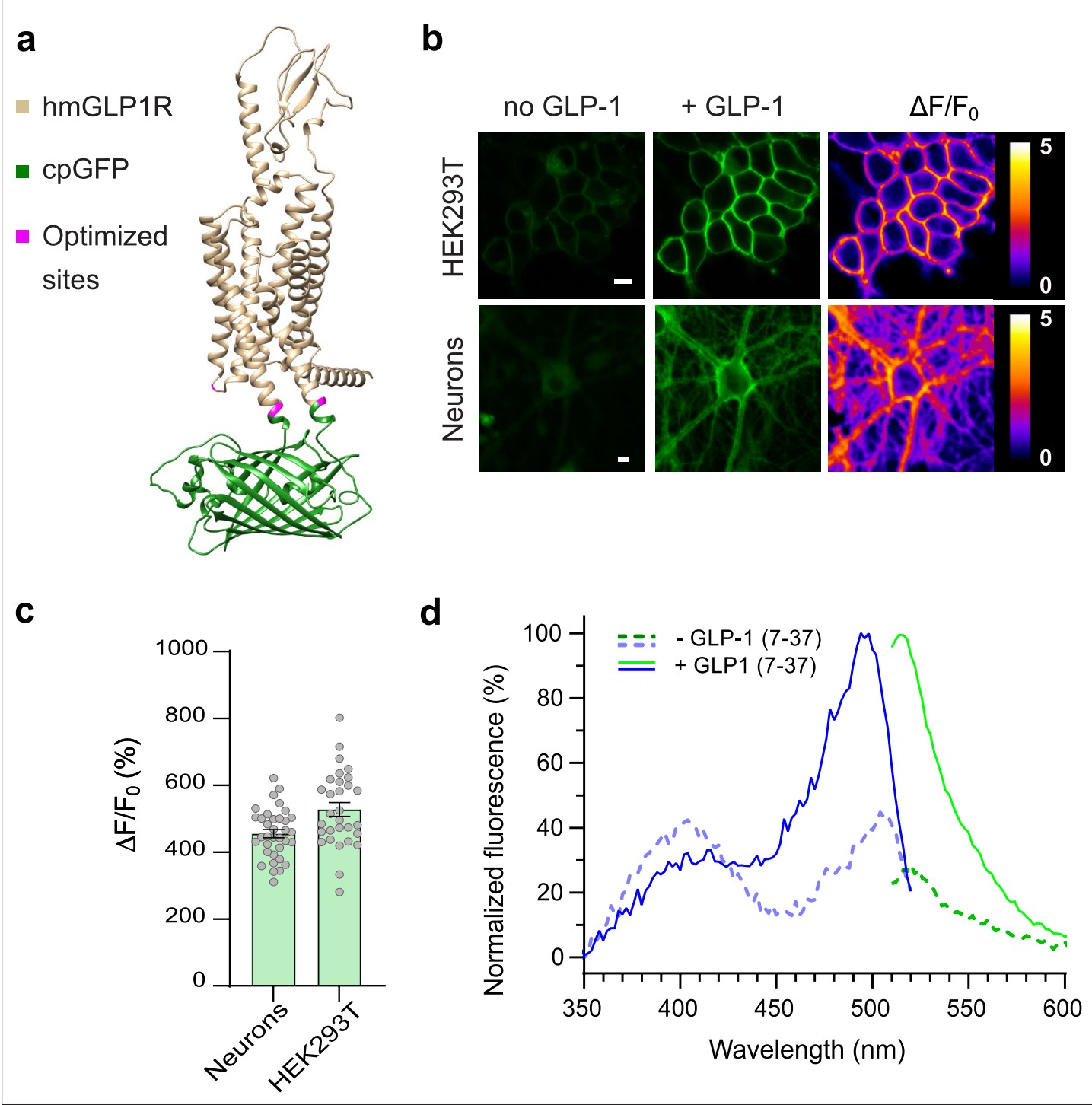

**Figure 1.** Development and optical properties of GLPLight1. (**a**) Structural model of GLPLight1 obtained using Alphafold (*Mirdita et al., 2022*). The human glucagon-like peptide-1 receptor (GLP1R) is shown in gold, cpGFP in green, and residue targets of mutagenesis are shown in magenta. (**b**) Representative images showing GLPLight1 expression and fluorescence intensity change before (left) and after (center) addition of 10 μM glucagon-like peptide-1 (GLP-1) (7–37) as well as their respective pixel-wise $\Delta F/F_0$ images in HEK293T cells (top) and primary cortical neurons (bottom). Scale bars, 10 μm. (**c**) Maximal fluorescence response of GLPLight1 expressed in the indicated cell types after the addition of 10 μM GLP1. n=35 neurons and n=30 HEK293T cells, from 3 independent experiments. (**d**) One-photon excitation/emission spectra of GLPLight1-expressing HEK293T cells before (dark green and dark blue) and after (light green and light blue) addition of 10 μM GLP-1 (7–37) normalized to the peak excitation and emission of the GLP-1-bound state of the sensor. Data were obtained from three independent experiments. Only mean values are shown. All data shown as mean ± SEM unless stated otherwise.

*Figure 1 continued on next page*

*Figure 1 continued*

The online version of this article includes the following source data and figure supplement(s) for figure 1:

**Source data 1.** Development and optical properties of GLPLight1.

**Figure supplement 1.** Screening process for the development of GLPLight1.

**Figure supplement 1—source data 1.** Screening process for the development of GLPLight1.

**Figure supplement 2.** Development of the control sensor GLPLight-ctr.

**Figure supplement 2—source data 1.** Development of the control sensor GLPLight-ctr.

**Figure supplement 3.** Intracellular signaling characterization of glucagon-like peptide-1 receptor (GLP1R) and GLPLight1.

**Figure supplement 3—source data 1.** Intracellular signaling characterization of glucagon-like peptide-1 receptor (GLP1R) and GLPLight1.

S441A, S442A, S444A, S445A, and T448A) aiming to maximally reduce the possibility of sensor internalization. The resulting sensor variant showed good membrane expression and a 528% maximal fluorescence response upon GLP-1 binding (*Figure 1b–c*, *Figure 1—figure supplement 1h*). This final variant was named GLPLight1 and was chosen for further characterization. To aid with control experiments during GLPLight1 validation, we also set out to develop a sensor variant carrying mutations in the peptide binding pocket. We screened a panel of 14 single point mutations and identified a combination of 3 mutations (L141A, N300A, and E387A) that abolished the fluorescent response to GLP-1 application while showing a good membrane expression of the sensor (*Figure 1—figure supplement 2a–c*). This control variant was named GLPLight-ctr.

## In vitro characterization of GLPLight1

To establish the utility of GLPLight1 as a new tool to investigate the hmGLP1R in pharmacological assays, we first characterized its properties in vitro. We started by comparing sensor expression and fluorescent response among different cell types. To do so, we expressed GLPLight1 in primary cortical neurons in culture, via adeno-associated virus (AAV) transduction. Two weeks post-transduction, GLPLight1 was well expressed on the neuronal membrane and showed a maximal response of 456% to GLP-1 application (10 µM) (*Figure 1b–c*). We then measured the spectral properties of the sensor in HEK cells. The fluorescence spectra were similar to those of previously described green GPCR-sensors (*Duffet et al., 2022a*; *Sun et al., 2018*), and showed a peak excitation around 500 nm, peak emission around 512 nm, and an isosbestic point at around 425 nm (*Figure 1d*). Work on previously developed GPCR-based sensors that respond to neuropeptide ligands (*Duffet et al., 2022a*; *Ino et al., 2022*) revealed that the conformational activation kinetics of these receptor types is at least an order of magnitude slower than what has been reported for monoamine receptors (*Feng et al., 2019*; *Patriarchi et al., 2018*; *Sun et al., 2018*; *Wan et al., 2021*), likely reflecting the more complex and polytopic binding mode of peptide ligands to their receptor.

Next, we compared the coupling of GLPLight1 and its parent receptor (WT GLP1R) to downstream signaling. We first measured the agonist-induced membrane recruitment of cytosolic miniG proteins and β-arrestin-2 using a split nanoluciferase complementation assay (*Dixon et al., 2016*). In this assay, both the sensor/receptor and the miniG proteins contain part of a functional luciferase (smBit on the sensor/receptor and LgBit for miniG proteins) that becomes active only when these two partners are in close proximity (*Wan et al., 2018*). In agreement with the known pleiotropic signaling of WT GLP1R (*Rowlands et al., 2018*), in our assay activation of the receptor led to a strong recruitment of miniGs, miniGq, miniGi, β-arrestin-2, as well as miniG12, albeit to a lower extent. In comparison to WT GLP1R, the coupling of GLPLight1 to all tested signaling partners was significantly reduced (*Figure 1—figure supplement 3a–j*). To further confirm the absence of coupling to intracellular cyclic-AMP (cAMP) signaling of GLPLight1, we performed a titration of GLP-1 on the sensor and WT GLP1R in a luminescence-based cAMP assay. This revealed that the WT GLP1R showed could potently elicit intracellular cAMP increases with an $EC_{50}$ of 8.0 pM whereas no such increase was observed for GLPLight1 even at the highest GLP-1 concentrations tested (100 nM, *Figure 1—figure supplement 3k*). We also performed a titration of GLP-1-induced recruitment of miniGs protein where we could show that GLP1R effectively recruits miniGs proteins with an $EC_{50}$ of 3.8 nM (*Figure 1—figure supplement 3l*). These results indicate that GLPLight1 is unlikely to couple with endogenous intracellular signaling pathways.

## Application of GLPLight1 as a tool for pharmacological screening

GLPLight1 is a novel genetically encoded sensor capable of providing a sensitive intensiometric readout of hmGLP1R activation in response to its endogenous ligands. As such, this tool could have great potential for applications in the drug discovery and development field; however, a more careful characterization of its pharmacological response profile is needed to ensure its implementation as a screening tool. We thus performed a series of in vitro pharmacological experiments in which we characterized GLPLight1 responses under different conditions and with a variety of ligands with known pharmacological effects on GLP1R, with the aim to demonstrate the applicability of this sensor as a pharmacological screening tool. We started by testing the reversibility of sensor response via competition of GLP-1 with an antagonist peptide. To do so, we imaged GLPLight1-expressing HEK293T cells upon addition, in sequence, of 1.0 μM GLP-1 followed by 10 μM exendin-9 (Ex-9), a well-known peptide antagonist of GLP1R. Ex-9 could partially reverse the signal to 42% of the maximal GLP-1 response, within less than 5 min in vitro (*Figure 2a–b*). Next, we tested whether two clinically used anti-obesity drugs that are known GLP1RAs, liraglutide or semaglutide (*O'Neil et al., 2018*), could trigger a response from the sensor. As expected, GLPLight1 responded to both GLP1RAs with almost maximal activation, on par with GLP1 (*Figure 2a*). These results indicate that GLPLight1 can serve as a direct readout of pharmacological drug action on the hmGLP1R with higher temporal resolution than previously available approaches, such as downstream signaling assays (*Zhang et al., 2020*).

Knowing that GLP-1 is produced along with GLP-2 and glucagon via proteolytic processing of a common PPG precursor protein (*Drucker, 2001*), we decided to investigate the specificity of our sensor against these other peptides. While the sensor did not respond with any detectable increase in fluorescence to GLP-2, it responded to glucagon with a $\Delta F/F_0$ of 324% (61% of maximal response to GLP-1). To further characterize the sensitivity of GLPLight1 to its two endogenous agonists, we performed titrations of GLP-1 and glucagon in HEK293T cells and determined that GLPLight1 had a 94-fold higher affinity for GLP-1 compared to glucagon ($EC_{50}$=28 nM for GLP-1, $EC_{50}$=2.6 μM for glucagon), in agreement with previously reported results employing a downstream cAMP readout (*Runge et al., 2003*). Furthermore, the affinity of GLP-1 measured in primary neurons ($EC_{50}$=9.3 nM) was comparable to the one in HEK cells (*Figure 2c*). Additionally, GLPLight1 did not respond to a panel of other endogenous class B1 GPCR peptide ligands that were tested at high concentration (1.0 μM), including GIP, CRF, PTH, PACAP, or VIP.

The binding of GLP-1 to its receptor occurs via the N-terminus of the peptide, as demonstrated by previous structural (*Jazayeri et al., 2017*) and mutagenesis studies (*Longwell et al., 2021*; *Zhang et al., 2020*). We therefore set out to determine whether the general trends observed by fluorescence response of GLPLight1 is in agreement with the pharmacological readout of GLP1R activation obtained using classical assays (*Adelhorst et al., 1994*). We synthesized four single-residue alanine mutants of GLP-1 at selected N-terminal positions (H1A, E3A, G4A, T5A) using automated fast-flow peptide synthesis (AFPS, see Supplementary Information) (*Hartrampf et al., 2020*; *Mijalis et al., 2017*). All peptides were obtained in good yields and excellent purities after RP-HPLC purification. Titrations of individual GLP-1 mutants on GLPLight1-expressing cells revealed clear effects of the mutations on either the maximal sensor response ($E_{max}$), the potency ($EC_{50}$) of the peptide ligand, or both (*Table 1* and *Figure 2d*). In particular, the critical role of H1 and G4 for both binding and activation has been reported in the literature several times (*Manandhar and Ahn, 2015*). In agreement with these results, we observed a significant reduction of $E_{max}$ and $EC_{50}$ for H1A (56% and 1300 nM, respectively) (*Table 1*, Entry a) and G4A (14% and 993 nM, respectively) (*Table 1*, Entry c), compared to WT GLP-1 using GLP1Light as a readout. Furthermore, position E3 was reported to be critical for binding, but not for activation. Here, we determined an $E_{max}$ of 96% compared to WT GLP-1 (*Table 1*, Entry b), as well as a reduced $EC_{50}$ (757 nM) for E3A, which is in agreement with the literature (*Manandhar and Ahn, 2015*; *Table 1*). Finally, T5 has been reported as less important for GLP1R binding and activation than the other investigated peptide positions (*Adelhorst et al., 1994*). Accordingly, our experiments with GLPLight1 T5A showed the highest $E_{max}$ (100%) and $EC_{50}$ (188 nM) (*Table 1*, Entry d) of all alanine mutants investigated herein. Overall, we conclude that fluorescence response of GLPLight1 can be used to study the relative trends for $E_{max}$ and potency of GLP1R ligands.

State-of-the-art techniques for detecting endogenous GLP-1 or glucagon release in vitro from cultured cells or tissues consist of costly and time-consuming antibody-based assays (*Kuhre et al., 2016*) or analytical chemistry procedures (*Amao et al., 2015*). Given the genetically encoded nature

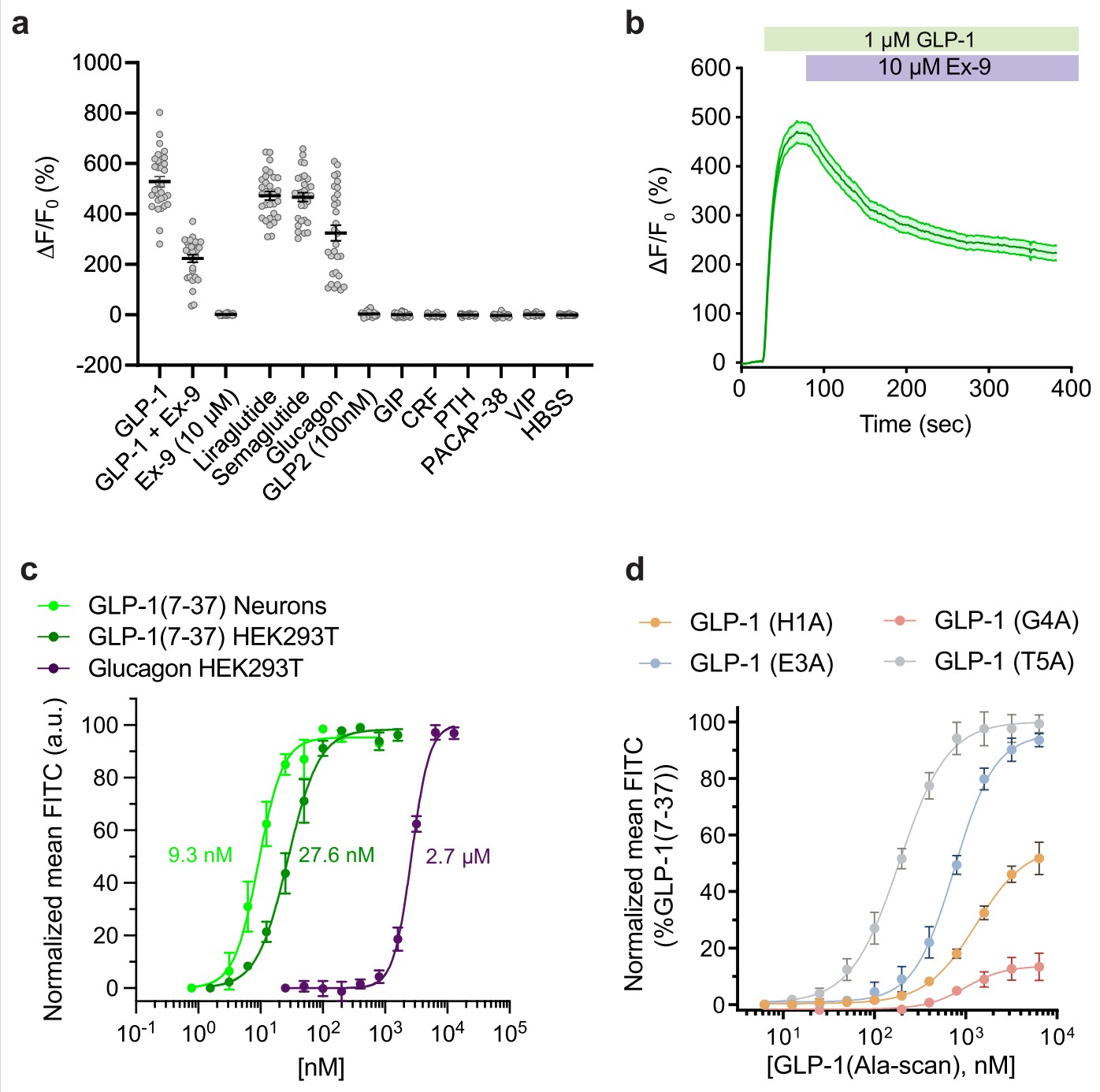

**Figure 2.** Pharmacological characterization of GLPLight1. (**a**) Absolute $\Delta F/F_0$ responses of GLPLight1 expressing HEK293T cells to various glucagon-like peptide-1 (GLP-1) agonists, antagonist, or to other class B1 neuropeptide ligands applied at 1 μM final (unless stated otherwise). n=30 cells from three independent experiments for all conditions. (**b**) $\Delta F/F_0$ responses from timelapse imaging experiments in which 1 μM GLP-1 and 10 μM exendin-9 (Ex-9, a peptide antagonist of glucagon-like peptide-1 receptor [GLP1R]) were subsequently bath-applied onto GLPLight1-expressing HEK293T cells. n=30 cells from three independent experiments. (**c**) Normalized dose-response curves showing the fluorescent responses of GLPLight1-expressing HEK293T cells and primary cortical neurons to GLP-1 (dark green and light green, respectively) or GLPLight1-expressing HEK293T cells to glucagon (purple). The curves fit were performed using a four-parameter equation and the mean $EC_{50}$ values determined are shown next to the traces in the corresponding color. n=3, 6 and 3 independent experiments for GLP-1 (7–37) in neurons, GLP-1 (7–37) and glucagon in HEK293T cells, respectively. (**d**) Dose-response curves showing the fluorescent responses of GLPLight1-expressing HEK293T cells to alanine mutants of the GLP-1 peptide normalized to the maximum mean fluorescence response (FITC intensity) obtained for the WT GLP-1 peptide. n=3 independent experiments for each peptide. All data are displayed as mean ± SEM.

*Figure 2 continued on next page*

*Figure 2 continued*

The online version of this article includes the following source data and figure supplement(s) for figure 2:

**Source data 1.** Pharmacological characterization of GLPLight1.

**Figure supplement 1.** Detecting endogenous glucagon-like peptide-1 (GLP-1) release from enterocrine cells using GLPLight1.

**Figure supplement 1—source data 1.** Detecting endogenous glucagon-like peptide-1 (GLP-1) release from enterocrine cells using GLPLight1.

and the fast optical readout of GLPLight1, this tool has the potential to facilitate studies investigating the physiological regulation of GLP-1 release in vitro. To establish whether GLPLight1 could be sensitive enough to detect endogenous GLP-1 release in an in vitro setting, we cultured sensor-expressing HEK293T cells in the presence or absence of a GLP-1/glucagon-producing immortalized enteroendocrine cell line (GLUTag cells, *Brubaker et al., 1998*). To distinguish the two cell types in the co-culture system, the HEK239T cells were co-transfected with a cytosolic red fluorescent protein (mKate2). To detect whether the GLPLight-expressing cells had detected endogenous GLP-1 release by the ECs, we bath-applied GLP-1 to cause full activation of the sensor. We observed that the response to GLP-1 of sensor-expressing cells cultured in the presence of GLUTag cells was significantly lower than that of cells cultured in their absence (*Figure 2—figure supplement 1*). These results indicate that GLPLight1 was partially pre-activated by endogenous GLP-1 secreted by the ECs present in the dish. The detection of endogenous GLP-1 by the sensor opens the possibility to use it as a screening tool for studying intrinsic/extrinsic factors that regulate GLP-1 release from ECs in vitro.

## Development and in vitro characterization of photo-GLP1

To investigate the spatiotemporal activation of GLP1R and GLPLight1, a photocaged derivative of GLP-1 was envisioned. To ensure that the photo-GLP1 does not activate GLP1R or GLPLight1 prior to uncaging (i.e. in the dark), the photocage must be located on or near GLP-1 regions that are essential for binding. Photocaging of peptides can be achieved by the attachment of a photocaging molecule at a side-chain functionality, backbone amide, or at the C- or N-terminus of the peptide. Recently, we reported the optical control of orexin-B using a UV-visible light-sensitive C-terminal photocage (*Duffet et al., 2022b*). As opposed to orexin-B, GLP-1 primarily binds via its N-terminus to GLP1R (*Jazayeri et al., 2017*). We therefore explored an N-terminal caging strategy to generate a photo-GLP1 (*Figure 3a*). GLP-1 was prepared by solid-phase peptide synthesis utilizing AFPS (*Hartrampf et al., 2020*; *Mijalis et al., 2017*). Before cleavage of the peptide from the resin, photocaging of the GLP-1 N-terminal amine was carried out by treating the resin-bound peptide with an active ester (*N*-hydroxysuccinimide ester) form of the nitrobenzene-type photocage (see *Source data 1*). Cleavage of the resulting photocaged peptide from the resin followed by RP-HPLC purification successfully provided photo-GLP1 in 5% overall yield with >95% purity. To confirm the release of WT GLP-1 upon treatment of photo-GLP1 with UV light, photo-GLP1 (80 µM in HBSS) was irradiated under LED light ($\lambda$ =370 nm, 0.64 mW/mm$^2$) for 20 min with air cooling. Subsequent LCMS and UHPLC analysis demonstrated complete uncaging of photo-GLP1 to afford WT GLP-1, confirmed by co-injection of a standard sample of WT GLP-1 (80 µM in HBSS) (*Figure 3—figure supplement 1*).

We then leveraged on GLPLight1 to establish an all-optical assay for characterizing photo-GLP1 uncaging in vitro. We bath-applied photo-GLP1 (10 µM) onto GLPLight1-expressing HEK293T cells

**Table 1.** Titration parameters of alanine scanned variants of glucagon-like peptide-1 (GLP-1) peptide.

The $E_{max}$ and $pEC_{50}$ values were derived from the four-parameter non-linear fit for each peptide and the $EC_{50}$ shift by comparison against WT GLP-1 peptide measured alongside.

| Entry | GLP-1 variant | $E_{max}$ (% WT GLP-1) | $pEC_{50}$ (M) | Fold-reduction $EC_{50}$ vs. WT GLP-1 |
|-------|---------------|------------------------|----------------|----------------------------------------|
| a | H1A | 56±4 | 5.89±0.05 | ≈ 63 |
| b | E3A | 96±3 | 6.10±0.03 | ≈ 37 |
| c | G4A | 14±3 | 6.00±0.13 | ≈ 48 |
| d | T5A | 100±4 | 6.73±0.04 | ≈ 9 |
| n/a | WT GLP-1 | 100 | 7.69±0.04 | 0 |

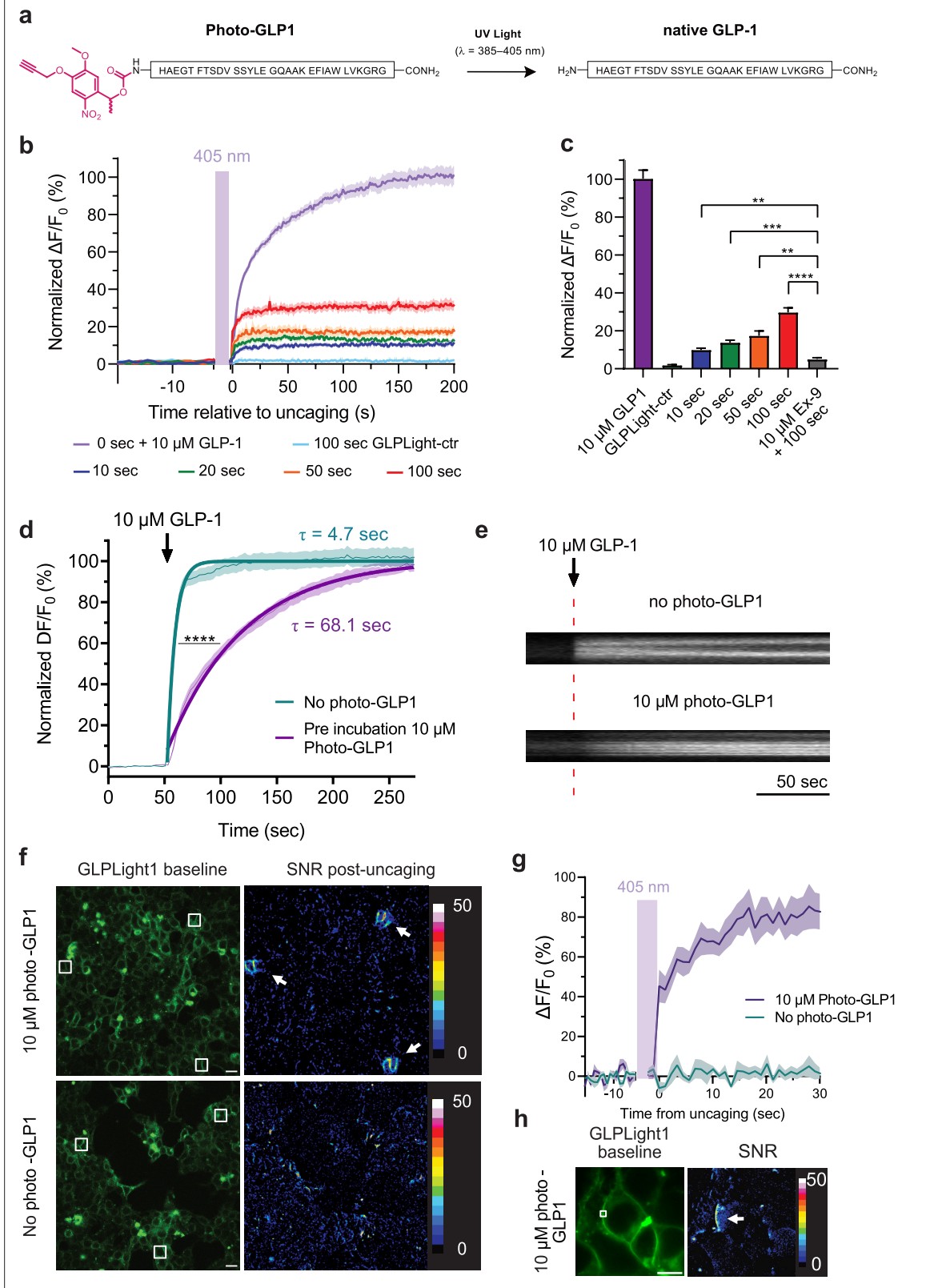

**Figure 3.** All-optical visualization and control of human glucagon-like peptide-1 receptor (GLP1R) activation. (**a**) Schematic representation of the N-terminal chemical caging strategy used to generate photocaged glucagon-like peptide-1 derivative (photo-GLP1). The peptide product (native GLP-1) after optical uncaging is shown. (**b**) Timelapses of fluorescence response for GLPLight1 or GLPLight-ctr expressing HEK293T cells before and after optical uncaging (purple vertical bar, 405 nm laser, scanning rate of 0.8 Hz and variable durations as specified below the graph). The values were

*Figure 3 continued on next page*

*Figure 3 continued*

normalized to the maximal response of GLPLight1 between t=150 and 200 s to 10 µM GLP-1 (purple trace). In all cases, cells were pre-incubated for 1–2 min with 10 µM photo-GLP1 before imaging and optical uncaging started. All fluorescent signals were analyzed within 20 µm distance from the uncaging area. n=11–19 cells from three independent experiments. (**c**) Quantification of the normalized average fluorescence from (**b**) between t=25–75 s for uncaging experiments and t=150–200 s for GLP-1 application experiments. All uncaging experiments on GLPLight1 were compared to the one with pre-incubation with exendin-9 (Ex-9, see *Figure 3—figure supplement 2*) using Brown-Forsythe ANOVA test followed by Dunnett's T3 multiple comparison. p=0.0061; 0.0001; 0.0026 and 1293×10$^{-6}$ for 10, 20, 50, and 100 s uncaging events, respectively. (**d**) Fluorescence response of GLPLight1-expressing HEL293T cells to 10 µM GLP-1 either after pre-incubation with 10 µM photo-GLP1 (magenta) or in the absence of it (blue). The data were normalized to the maximal response of GLPLight1-expressing cells in the absence of photo-GLP1 and fitted with a non-linear mono exponential fit to determine $\tau$ values. Statistical analysis was performed using the extra sum-of-squares F test; ****p<0.0001; n=18 and 17 cells from three independent experiments in the absence or presence of photo-GLP1 respectively. All data are displayed as mean ± SEM. (**e**) Kymographs of representative cells from (**d**) showing the fluorescence intensity of a line drawn across a cell membrane over time in the absence (top) or presence (bottom) of photo-GLP1 (10 µM) in the bath. The timepoint of GLP-1 application is shown by the red dotted line. (**f**) Representative images of multiple uncaging events performed at different locations across the field of view. Images show the basal fluorescence of GLPLight1-expressing HEK cells (left) in the presence (top) or absence (bottom) of 10 µM photo-GLP1, as well as the corresponding pixel-wise heatmap of SNR post-uncaging. Localized functional sensor responses to optical uncaging of photo-GLP1 are indicated by white arrows. Uncaging was performed for a duration of 40 s in total for all the three areas shown as white squares using a scanning rate of 1.5 Hz. Scale bars: 20 µm. (**g**) Quantification of the timelapse of fluorescence response of GLPLight1 from (**f**) inside the uncaging areas. (**h**) Same as (**f**) but with a sub-cellular localized uncaging region selected on the membrane of a GLPLight1-expressing cell with 1.5 s uncaging duration and a 25 Hz scanning rate. Scale bar 10 µm.

The online version of this article includes the following source data and figure supplement(s) for figure 3:

**Source data 1.** All-optical visualization and control of human glucagon-like peptide-1 receptor (GLP1R) activation.

**Figure supplement 1.** Biochemical characterization of photocaged glucagon-like peptide-1 derivative (photo-GLP1).

**Figure supplement 1—source data 1.** Biochemical characterization of photocaged glucagon-like peptide-1 derivative (photo-GLP1).

**Figure supplement 2.** Further characterization of photocaged glucagon-like peptide-1 derivative (photo-GLP1) uncaging.

**Figure supplement 2—source data 1.** Further characterization of photocaged glucagon-like peptide-1 derivative (photo-GLP1) uncaging.

and performed optical uncaging by exposing a defined area directly next to the cells to 405 nm laser light (UV light) for defined periods of time, while the sensor fluorescence was imaged using 488 nm laser light. Application of photo-GLP1 by itself failed to trigger any response from GLPLight1, indicating a lack of functional activity in the absence of UV light (*Figure 3—figure supplement 2a*). On the contrary, after photo-GLP1 was added to the bath, the fluorescence of GLPLight1 visibly increased upon 10 s of UV light exposure, indicating that GLP-1 could successfully be uncaged and activated the sensor on the cells. Higher durations of UV light exposure led to a higher degree of GLPLight1 responses, and the maximal uncaging duration tested (100 s) triggered approximately 30% of the maximal response of the sensor, as assessed in the same assay by bath application of a saturating GLP-1 concentration (10 µM) (*Figure 3b–c*). Importantly, to show that the sensor signals are not due to UV light-induced artifacts, we reproduced the maximal (100 s) uncaging protocol on GLPLight-ctr-expressing HEK293T cells and confirmed that in this case no sensor response could be observed. Furthermore, pre-treatment of the cells with the GLP1R antagonist Ex-9 significantly blunted the sensor response evoked by the optical uncaging (100 s) (*Figure 3c*, *Figure 3—figure supplement 2b*). These results indicate that photo-GLP1 can be effectively uncaged in vitro using 405 nm light to control hmGLP1R activation.

## High-resolution all-optical visualization and control of GLP1R activity

Upon performing the uncaging experiments, we noticed that the profile of the sensor response to bath-applied GLP-1 differed, depending on whether or not photo-GLP1 was present in the bath. To investigate this phenomenon more in detail, we measured and compared the sensor activation kinetics when GLPLight1 was activated by direct bath application of GLP-1 in the presence or absence of an equimolar concentration of photo-GLP1 in the bath. The sensor response was strikingly different in the two conditions, and exhibited an approximate 14-fold reduction in the speed of activation in the presence of photo-GLP1 ($\tau_{ON}$ without photo-GLP1=4.7 s; $\tau_{ON}$ with photo-GLP1=68.1 s; *Figure 3d–e*). These results indicate that photo-GLP1, in the dark (i.e. with an intact photocage), can affect the kinetics of GLP1R activation, and this is likely mediated by its binding to the receptor extracellular domain (ECD), which competes for the functionally active GLP-1. In fact, since the GLP1R belongs to class B1 GPCRs, the binding of GLP-1 is known to involve an initial step where the peptide

C-terminus is recruited to the ECD, followed by a second step involving insertion of the peptide N-terminus into the receptor binding pocket (**Wu et al., 2020**). Given that our photocage was placed at the very N-terminus of photo-GLP1, our results show that this caging approach prevents the peptide's ability to activate GLPLight1 but, at the same time, preserves its ability to interact with the ECD.

We next asked whether we could leverage GLPLight1 to obtain spatial information on the extent of GLP1R activation in response to photo-GLP1 uncaging. To do so, we performed optical photo-GLP1 uncaging on three separate areas of about 400 $\mu m^2$ placed at different locations in a large field of view (FOV, approximately 40,000 $\mu m^2$). UV light was applied for a total of 40 s on the three uncaging regions during the imaging session. GLPLight1 shows a fluorescent response in all three uncaged areas, while its fluorescence remained unaltered throughout the rest of the FOV, indicating high spatial localization of the response to GLP-1 (**Figure 3f**). As a control, the omission of photo-GLP1 in the cell bath led to no sensor response upon uncaging (**Figure 3g**). Additionally, the same session was repeated on GLPLight-ctr-expressing cells. Also in this case, no response to uncaging could be observed (**Figure 3f**). To determine whether the sensor readout in this assay could report GLP1R activation with even sub-cellular resolution, we repeated the uncaging experiment by selecting an uncaging area of approximately 16 $\mu m^2$ directly on a cell membrane. In this case, the application of UV light led to localized activation of GLPLight1 that was limited to a portion of the cell membrane and did not spread to neighboring cells (**Figure 3h**). These results demonstrate that the optical nature of the GLPLight1 readout makes it possible to determine the spatial extent of GLP1R activation with very spatial high-resolution, down to sub-cellular domains.

Finally, we tested whether uncaging of photo-GLP1 could be used to control functional signaling downstream of hmGLP1R activation. To this aim, we employed a recently developed genetically encoded sensor for cAMP (G-Flamp1) (**Wang et al., 2022**), which is the main second messenger involved in cellular signaling downstream of GLP1R activation (**Holz et al., 2015**). We imaged a field of HEK293T cells co-transfected with the hmGLP1R and G-Flamp1 (**Figure 4a**) during application of photo-GLP1 to the cells and after optical uncaging of photo-GLP1 (2 s, 1 nM) within a limited area of about 70 $\mu m^2$ located directly above a single HEK293T cell. As a result of uncaging, the signal from the cAMP sensor increased visibly and significantly only in the cell directly underneath the uncaged area (**Figure 4b**). The same uncaging protocol applied in the absence of photo-GLP1 on the cells failed to trigger any response from the cAMP sensor , indicating that the sensor signals reliably reported intracellular cAMP signaling triggered by uncaged GLP-1. Furthermore, as a positive control, we bath-applied the same concentration of GLP-1 (1 nM) at the end of each recording to stimulate simultaneously the activation of the receptor on all cells. Indeed, this could elicit a response in all the imaged cells that did not respond to the previous uncaging protocol ('distant cells') (**Figure 4b–d**). As part of our observations, we observed a small dip of the G-Flamp-1 signal in response to photo-GLP1 bath application (**Figure 4—figure supplement 1**). To assess whether this signal drop was caused by the signaling activity of the photo-GLP1 or was an artifact from G-Flamp-1 imaging, we repeated the measurement by applying HBSS to the cells. The small signal drop could be detected also in these experiments (**Figure 4—figure supplement 1**), demonstrating that the initial dip in G-Flamp-1 signal was artefactual, possibly due to temperature or pressure changes onto the cells. Overall, our results demonstrate that uncaging of photo-GLP1 can be used to achieve optical control of GLP1R signaling activation with high spatiotemporal resolution.

## Discussion

Here, we report the first genetically encoded sensor engineered based on cpGFP and the hmGLP1R. We show that this tool can directly report ligand-induced conformational activation of this receptor with the high sensitivity and spatiotemporal resolution typical of GPCR-based sensors. Using this new probe, we found that ligand-induced conformational activation of the hmGLP1R occurs on slower timescales compared to the reported kinetics of other similarly built GPCR sensors (**Labouesse and Patriarchi, 2021**). This new insight is not surprising given that previously developed sensors were built from class A GPCRs (**Labouesse and Patriarchi, 2021**), while GLP1R belongs to a different class of GPCRs (class B1) that is characterized by a distinct ligand-binding mechanism that involved initial ligand 'capture' by the receptor's ECD, followed by ligand insertion into the receptor binding pocket for initiating the transduction of signaling (**Zhang et al., 2020**). As a reference, other previously characterized class A GPCR-based neuropeptide biosensors showed sub-second activation kinetics

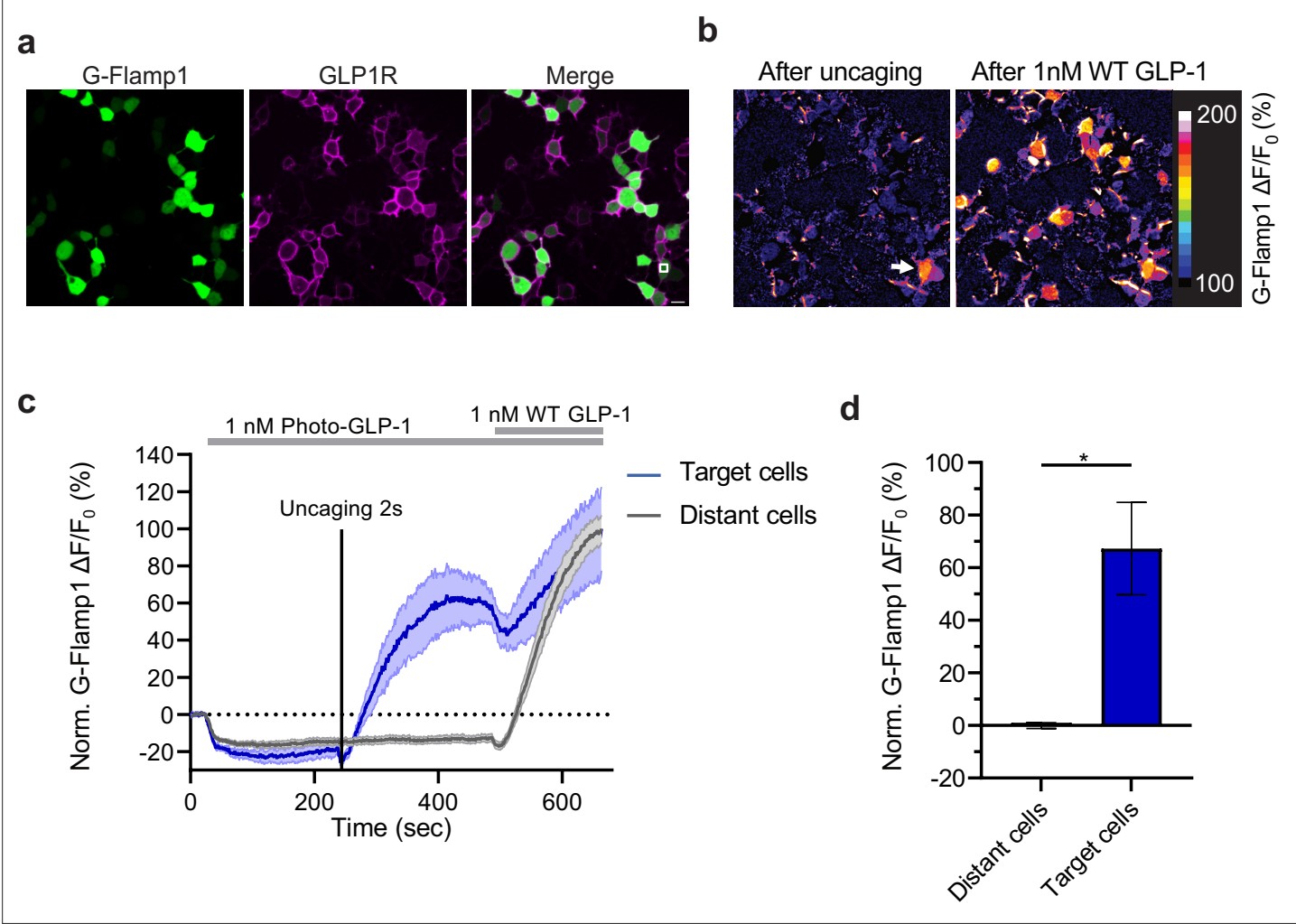

**Figure 4.** Effect of photocaged glucagon-like peptide-1 derivative (photo-GLP1) uncaging on intracellular signaling. (**a**) Representative images of HEK293T cells used in (**c,d**). Human glucagon-like peptide-1 receptor (hmGLP1R) expression was visualized using an Alexa-647-conjugated anti-FLAG antibody. The uncaging region is represented by the white square in the lower right area. (**b**) Representative images from the pixel-wise fluorescence response from G-Flamp1 after uncaging (left) and bath application of 1 nM WT GLP-1 (right). The white arrow indicates the localized area of uncaging. (**c**) Fluorescence change during timelapse imaging of G-Flamp1/GLP1R co-expressing cells after addition of 1 nM photo-GLP1 and localized uncaging (405 nm, 2 s, and 32 Hz scanning rate), followed by bath application of 1 nM WT GLP-1. The timepoints of ligand addition are represented using gray rectangles and the uncaging bout by the vertical black line. Quantification of the fluorescence response is shown separately for 'target cells' (blue, cells within the uncaging area) and for 'distant cells' (gray, cells positioned at least 10 μm away from the uncaging area). The fluorescent responses from G-Flamp1 were normalized to the maximal activation after addition of WT GLP-1. (**d**) Quantification from the average normalized fluorescence in (**c**) between t=400 and t=450 s using the 10 frames before uncaging as a baseline for each condition. n=3 'target cells' and 15 'distant cells' from three independent experiments. *p=0.03061 using a one-tailed Student's t-test with Welch's correction. All scale bars: 20 μm and all data displayed as mean ± SEM.

The online version of this article includes the following source data and figure supplement(s) for figure 4:

**Source data 1.** Effect of photocaged glucagon-like peptide-1 derivative (photo-GLP1) uncaging on intracellular signaling.

**Figure supplement 1.** Control experiment for intracellular signaling characterization.

**Figure supplement 1—source data 1.** Control experiment for intracellular signaling characterization.

(**Duffet et al., 2022a**; **Ino et al., 2022**). Accordingly, our observations show that the receptor activation kinetics can be largely influenced by pre-incubation with an inactive form of the GLP-1 peptide (photo-GLP1), likely because the inactive peptide interacts with and occupies the receptor's ECD.

We showcased the sensitivity and utility of GLPLight1 as a pharmacological tool to aid drug screening and development efforts by characterizing its response to various naturally occurring

peptide ligands, as well as clinically used agonists and peptide derivatives with diverse pharmacological actions on GLP1R. Besides its applications in pharmacology and drug discovery, given the high sensitivity and lack of interference with intracellular signaling of GLPLight1, it might be possible to employ this tool to investigate the dynamics of endogenous GLP-1 and/or glucagon directly in living systems (in vivo), although based on the evidence provided in this study the in vivo utilization of the sensor is not guaranteed to succeed.

The apparent EC$_{50}$ of GLPLight1 fluorescence response to GLP-1 is very similar to that measured for miniGs recruitment to the hmGLP1R, while it is approximately three orders of magnitude lower than that of the cAMP response downstream of hmGLP1R. This discrepancy might be due to intrinsic differences of the assays used or to intrinsic differences in the distinct aspects of the signaling pathway investigated (i.e. direct recruitment of miniGs versus enzymatically amplified cAMP signals). This raises the interesting possibility that under physiological conditions GLP-1 might elicit different functional responses based on the location of its action and on the spatial concentration gradients on target cells/tissues.

Given that GLPLight1 produces a fluorescence readout that is more representative in terms of sensitivity to that measured by direct recruitment of miniGs proteins to the hmGLP1R, the characteristics of this sensor appear not suitable to detect the concentration range achieved by GLP-1 in the periphery through endocrine signaling (picomolar levels). Nevertheless, it is conceivable that under specific circumstances, for example in specific brain areas or in close proximity to enteroendocrine cells in the gut, levels of GLP-1 release might reach high-enough levels that could be detected by GLPLight1. Future studies could attempt in vivo use of the sensor to further explore this interesting direction, for example by leveraging on AAV-mediated expression of GLPLight1 in living tissues or animals for implementing its use through in vivo imaging techniques, such as fiber photometry (*Gunaydin et al., 2014*), mesoscopy (*Cardin et al., 2020*), or two-photon microscopy (*Helmchen, 2009*). Through such efforts, GLPLight1 might be helpful to shine new light on the hidden mechanisms of GLP-1 and/or glucagon release dynamics in relation to physiological or pathological conditions.

Finally, we leveraged GLPLight1 to characterize the uncaging of the photo-GLP1 described for the first time in this work. Optical tools to selectively activate GLP1R could contribute to mechanistic studies (*Chen et al., 2022*; *Frank et al., 2018*), and the photoswitchable GLP-1 LirAzo was recently used to optically control insulin secretion and cell survival (*Broichhagen et al., 2015*). As opposed to photoswitchable peptides, in which the side chain or part of the peptide backbone is replaced by a photoswitchable moiety such as an azobenzene, photo-GLP1 releases native GLP-1 upon optical uncaging. A drawback of a photocaging strategy, on the other hand, is that it is an irreversible transformation, unlike photoswitchable derivatives. By deploying GLPLight1 and photo-GLP1 in concert in an all-optical assay, we determined that the spatial spread of GLP1R activation in response to GLP-1 release can be localized to single-cells or even sub-cellular domains. Furthermore, by combining a state-of-the-art cAMP sensor with photo-GLP1, we demonstrated the optical control of hmGLP1R-dependent downstream cellular signaling with single-cell resolution, opening exciting new opportunities for investigating the spatial regulation of this signaling pathway. Since we photocaged native GLP1, it is important to note that the photo-GLP1 might still be susceptible to DPPIV-mediated degradation when used in in vivo applications. We envisage that our photo-GLP1 will nonetheless find applications in neurobiological in vivo studies in brain tissue, as DPPIV levels in the brain are significantly lower than in peripheral organs.

In summary, we developed and utilized a new all-optical toolkit to unveil a previously inaccessible spatial dimension of the GLP-1/GLP1R system. These tools may thus be readily implemented in a variety of applications, some of which are showcased as part of this study, to advance our understanding of the roles of GLP-1/glucagon/GLP1R signaling system in physiology, or to foster the drug screening and development process targeting the GLP1R pathway.

## Materials and methods

**Key resources table**

| Reagent type (species) or resource | Designation | Source or reference | Identifiers | Additional information |
|---|---|---|---|---|
| Gene (human) | GLP1R | Integrated DNA Technologies | | |
| | *Continued on next page* | | | |

| Reagent type (species) or resource | Designation | Source or reference | Identifiers | Additional information |
|---|---|---|---|---|
| Strain, strain background (*Escherichia coli*) | NEB 10-beta Competent *E. coli* | NEB | C3019 | |
| Commercial assay or kit | NEBuilder HiFi | NEB | E2621 | |
| Recombinant DNA reagent | Pfu-Ultra II Fusion | Agilent | 600387 | |
| Cell line (*Mus musculus*) | GLUTag enterocrine cell line | Daniel J Drucker (Univ. of Toronto) | | |
| Cell line (*Homo sapiens*) | Human Embryonic Kidney (HEK293T) | ATCC | CRL-3216 | |
| Antibody | Anti-FLAG-Alexa-647 | M1 (Sigma-Aldrich) In-house conjugated to Alexa-647 (Miriam Stoeber, University of Geneva) | F3040 | IF(1:1000) |
| Chemical compound | Fmoc- and side chain-protected L-amino acids | Bachem AG | | |
| Chemical compound | Piperidine | Chemie Brunschwig AG | 110-89-4 | 99% |
| Chemical compound | *O*-(7-Azabenzotriazol-1-yl)-*N,N,N′,N′*-tetramethyluronium hexafluorophosphate (HATU) | Bachem AG | 148893-10-1 | |
| Chemical compound | (7-Azabenzotriazol-1-yloxy)tripyrrolidinophosphonium hexafluorophosphate (PyAOP) | Advanced ChemTech CreoSalus | 156311-83-0 | |
| Chemical compound | *N,N*-Diisopropylethylamine (DIPEA) | Sigma-Aldrich Chemie GmbH | 7087-68-5 | |
| Chemical compound | Trifluoroacetic acid (TFA) | Sigma-Aldrich Chemie GmbH | 76-05-1 | For HPLC, ≥99.0% |
| Chemical compound | Triisopropylsilane (TIPS) | Sigma-Aldrich Chemie GmbH | 6485-79-6 | 98% |
| Chemical compound | 3,6-Dioxa-1,8-octane-dithiol (DODT) | Sigma-Aldrich Chemie GmbH | 14970-87-7 | 95% |
| Chemical compound | α-Methyl-5-methoxy-2-nitro-4-(2-propyn-1-yloxy)benzyl alcohol | Sigma-Aldrich Chemie GmbH | 1255792-05-2 | |
| Chemical compound | *N,N*'-Disuccinimidyl carbonate (DSC) | Sigma-Aldrich Chemie GmbH | 74124-79-1 | |
| Other | NovaPEG Rink Amide resins (0.41 mmol/g and 0.20 mmol/g loading) | Sigma-Aldrich Chemie GmbH | | |
| Other | AldraAmine trapping packets (volume 1000–4000 mL) | Sigma-Aldrich Chemie GmbH | | |

## Molecular cloning

The sequence coding for hmGLP1R was ordered as a synthetic DNA geneblock (Integrated DNA Technologies) bearing HindIII and NotI restriction site for cloning into a CMV-promoter plasmid (Addgene #60360). Sequences coding for the hemagglutinin secretion motif and a FLAG Tag were added to the N-terminus of the GLP1R open reading frame to increase plasma membrane expression and enable receptor labeling, respectively. Sensor variants were obtained using Gibson assembly (NEBuilder HiFi DNA Assembly Cloning Kit) (*Gibson et al., 2009*). Site-saturated mutagenesis was performed by PCR using primers bearing randomized codons at specified locations (NNK). For luminescence-based characterization of G protein and β-arrestin coupling, the small subunit (i.e. smBit) of NanoLuc (*Cannaert et al., 2016*) was PCR-amplified from a Beta2AR-SmBit donor plasmid and cloned at the C-terminal end of the GLP1R and GLPLight1 using Gibson assembly. PCRs were performed using a Pfu-Ultra II Fusion High Fidelity DNA Polymerase (Agilent). All sequences were verified using Sanger sequencing (Microsynth). For cloning GLPLight1 and GLPLight-ctr into the viral vector, BamHI and HindIII restriction sites were added flanking the sensor coding sequence by PCR amplification, followed by restriction cloning into pAAV-hSynapsin1-WPRE, obtained from the Viral Vector Facility of the University of Zürich.

## Structural modelling

The structural model of GLPLight1 was obtained using ColabFold (*Mirdita et al., 2022*) using pdb70 as a template mode. The best prediction was selected manually and edited using Chimera.

## Peptide synthesis and biochemical characterization

GLP-1, photo-GLP1, and all alanine scan peptides were synthesized on an AFPS using a recently developed protocol (*Hartrampf et al., 2020*). A detailed description of the synthetic procedures and all analytical data can be found in the Supplementary Information.

## Cell culture, imaging, and quantification

Mammalian HEK293T cells (CRL-3216 from ATCC) were authenticated by the vendor and tested negative for mycoplasma. They were cultured in DMEM medium (Thermo Fisher) supplemented with 10% FBS (Thermo Fisher) and 1× final Antibiotic-Antimicotic (Thermo Fisher) and incubated at 37°C in 5% $CO_2$. The cells were transfected using Effectene transfection kit for individual dishes or 24-well plates (QIAGEN) or Linear PEI (Sigma-Aldrich) for T75 flask transfection following the manufacturer's instructions and imaged 24–48 hr after transfection. GLUTag ECs were obtained indirectly from the laboratory what originally generated this cell line (*Drucker et al., 1994*). These cells were authenticated by the laboratory that originally generated them using mouse karyotyping and tested negative for mycoplasma. They were cultured on plates coated with 0.1% gelatine (Sigma-Aldrich) in low-glucose DMEM medium (1 g/L glucose) supplemented with L-glutamine (4 mM) and pyruvate (1 mM), 10% FBS and 1% Pen/Strep (Thermo Fisher). Primary cortical neurons were prepared as follows: the cerebral cortex of 18-day-old rat embryos were carefully dissected and washed with 5 mL sterile-filtered PBGA buffer (PBS containing 10 mM glucose, 1.0 mg/mL bovine serum albumin, and antibiotic-antimycotic 1:100 [10,000 units/mL penicillin; 10,000 µg/mL streptomycin; 25 µg/mL amphotericin B]) (Thermo Fisher Scientific). Cortices were cut into small pieces and digested in 5.0 mL sterile-filtered papain solution for 15 min at 37°C. Tissues were then washed with complete DMEM medium containing 10% fetal calf serum and penicillin/streptomycin (1:100), triturated and filtered through a 40 µm cell strainer. Neurons were plated at a concentration of 40,000–50,000 cells per well onto poly-L-lysine (50 µg/mL in PBS, Thermo Fisher Scientific) coated dishes and kept in NU-medium (Minimum Essential Medium with 15% NU serum, 2% B27 supplement, 15 mM HEPES, 0.45% glucose, 1.0 mM sodium pyruvate, 2.0 mM GlutaMAX). The cultures were virally transduced after 4–6 days with AAV at a $1×10^9$ GC/mL final titer and kept for 12–16 days in vitro. The HEK293T cells or neurons were rinsed with HBSS (Hank's Balanced Salt Solution, Life Technologies) and kept in a final volume of HBSS being either 100 µL for individual 15 mm glass bottom insert dish or 500 µL for 24-well plates. Timelapse recordings were performed at room temperature (approx. 20°C) on a Zeiss LSM 800 inverted confocal microscope controlled by Zeiss Zen Blue 2018 v2.6 software using either a 40× oil-based objective (individual dishes) or 20× air objective (24-well plates). The probes were excited using the following laser lines: 488 nm for GLPLight1 and GLPLight1-ctr. The ligands were all added in bolus before or during the timelapse recording using a micropipette to reach the desired final concentration once mixed with HBSS imaging media. Optical uncaging was performed using a 40× Plan-Apochromat oil-based objective (N/A=1.4; 69% transmittance at 405 nm from manufacturer's datasheet) over specified surface areas with various scanning rates (described in each legend) and a pixel dwell time of 1.52 µs. The average intensity of laser light used for uncaging was measured using an S120C Photodiode Power Sensor from Thorlabs and was kept at 0.38 mW. Image quantification was performed after manual selection of the regions of interest (ROI) corresponding to the cell membrane using the thresholding function from Fiji. The sensor response ($\Delta F/F_0$) was calculated as follows: $(F_t-F_0)/F_0$ with $F_t$ being the fluorescence intensity of the ROI at each timepoint t, and $F_0$ being the mean fluorescence intensity of the 10 timepoints before ligand addition for each ROI. $\Delta F/F_0$ values were calculated using a custom-made MatLab script and plotted in GraphPad Prism. The $\Delta F/F_0$ images were obtained by dividing pixel-wise fluorescence intensities prior and post ligand addition using a separate MatLab script and displayed as a color-coded RGB image. The custom-made MatLab scripts employed here have been described previously (*Duffet et al., 2022a*). They have been deposited on GitHub and are available for download at: https://github.com/PatriarchiLab/OxLight1.

## Plate reader-based imaging

The spectral characterization of the sensor was performed using GLPLight1 transfected HEK293T cells pre and post 10 µM GLP-1 (7–37) addition. The excitation and emission spectra were measured at $\lambda_{em}$ = 560 nm and $\lambda_{ex}$ = 470 nm, respectively, on a TECAN M200 Pro plate reader at 37°C. Transfected or untransfected cells were lifted using Versene (Thermo Fisher Scientific) and resuspended in PBS at

a concentration of 3.3 million cells per mL. For each condition, 300 µL of the cell suspension or PBS was transferred per individual wells of a black-bottom 96-well plate. Untransfected cells were used to correct for autofluorescence whereas PBS alone was used to subtract the buffer Raman bands. Intracellular cAMP production was assessed with the GloSensor cAMP assay. HEK293T cells were co-transfected with the pGLO20F and either hmGLP1Ror GLPLight1 in separate T75 flasks. Note that the endogenous signal peptide (amino acids 1–23) from GLP1R WT was deleted to maintain a similar membrane expression compared to GLPLight1 for all signaling assays. Cells were lifted 24 hr after transfection using Versene and re-suspended at a final concentration of 1,500,000 cells per mL in DMEM without phenol red+15 mM HEPES (Thermo Fisher Scientific). One-hundred µL of the cell suspension was dispensed per well in a 96-well white plate (Corning) and incubated with 2.0 mM of Luciferin potassium salt in 10 mM HEPES (pH 7.4) for 45–60 min. The cells were then imaged right after addition of 50 µL of ligand to reach the desired final concentration using a Cytation C10 (Biotek) plate reader in kinetic luminescence mode at 37°C. Positive (2.5 mM Forskolin) and negative controls (assay medium) were always included in triplicate alongside the constructs to be tested. The dose-response curves were obtained using the average luminescence value of the five timepoints after the peak of cAMP production of the positive control. Luminescence complementation assays were conducted using HEK293T cells co-transfected with GLPLight1-SmBit or GLP1R-SmBit along with either miniGs-LgBit, miniGi-LgBit, miniGq-LgBit, miniG12-LgBit, or Beta-arrestin-2-LgBit. After transfection, cells were seeded in a 96-well Optiplate, using 10,000 cells per well for the miniGs-LgBit condition and 50,000 cells for all others. Cells were then incubated for 45–60 min at 37°C with the NanoGlo live cell reagent according to the manufacturer's instructions. The baseline luminescence was recorded for 100 cycles (approx. 460 s), paused for manual addition of the ligand or the vehicle and resumed for another 200 cycles (approx. 920 s). The $\Delta R/R_0$ values were calculated by dividing the raw luminescence intensities after GLP-1 (7–37) addition by the ones after vehicle addition. This ratio was then normalized using the average luminescence intensity before addition as a baseline for both GLPLight1 and GLP1R conditions. The quantification of the maximal recruitment was calculated using the average $\Delta R/R_0$ between t=600 s and t=700 s for the timelapses and t=1600 s and t=1800 s for the GLP-1 titration of miniGs recruitment to GLP1R.

## Flow cytometry

After transfection, HEK293T cells were harvested using Versene. After resuspension in FACS buffer (1×PBS, 1.0 mM EDTA, 25 mM HEPES pH 7.0, 1% FBS) 300,000 cells were dispensed in each well of a 96-well plate, mixed with an equivalent volume of ligand to reach the desired concentration, and were incubated for 30 min at room temperature before the start of the measurement. Transduced neurons were washed once using the FACS buffer, gently mechanically lifted using a cell scraper and homogenized by repeated up and down-pipetting in FACS buffer. They were then incubated for 15 min on ice to minimize cell death before measurement. All flow cytometry experiments were performed on a FACS Canto II 2 L using a high-throughput sampler. Forward scattering, side-scattering, and 488 nm-excited fluorescence (FITC) data were acquired for a total of 50,000–100,000 events per well. The cells were manually gated using non-expressing cells as comparison, to define the FITC-positive population. Within this subgroup, the mean FITC intensity was calculated for each condition and normalized to the maximum FITC signal.

## Virus production

The AAV biosensors constructs used in this study were cloned by the Patriarchi laboratory. The VVF provided the backbone AAV constructs and produced the viruses. The titer of the viruses used were: AAVDJ.hSynapsin1.GLPLight1, $3.7\times10^{12}$ VG/mL; AAVDJ.hSynapsin1.GLPLight-ctr, $3.4\times10^{12}$ VG/mL.

## Animals

Animal procedures were performed in accordance with the guidelines of the European Community Council Directive or the Animal Welfare Ordinance (TSchV 455.1) of the Swiss Federal Food Safety and Veterinary Office and were approved by the Zürich Cantonal Veterinary Office (licence number: ZH087/2022). Rat embryos (E17) obtained from timed-pregnant Wistar rats (Envigo) were used for preparing primary cortical neuronal cultures.

## Statistical analyses

For in vitro analysis of sensor variants, where relevant the statistical significance of their responses was determined using a two-tailed unpaired Student's t-test with Welch's correction. For comparison of uncaging events in the presence or absence of antagonist statistical analysis was performed using Brown-Forsythe ANOVA test followed by Dunnett's T3 multiple comparison. For comparison of kinetic measurements, statistical analysis was performed using the extra sum-of-squares F test. All numbers of experimental repeats and p values are reported in the figure legends. Error bars represent mean ± standard error of the mean (SEM).

## Acknowledgements

The results are part of a project that has received funding from the European Research Council (ERC) under the European Union's Horizon 2020 research and innovation program (Grant agreement No. 891959) (TP). We also acknowledge funding from the University of Zürich and the Swiss National Science Foundation (Grant No. 310030_196455 and 310030L_212508) (TP) and (Grant No. 200021_200865) (NH). We would like to thank Jean-Charles Paterna and the Viral Vector Facility of the Neuroscience Center Zürich (ZNZ) for the help with virus production. All plasmids encoding miniG proteins-LgBit were a kind gift from Nevin A Lambert (University of Augusta). The plasmids encoding Beta2AR-SmBit and Beta-Arrestin-LgBit, as well as the Alexa-647-labeled M1 anti-FLAG antibody, were a kind gift from Miriam Stoeber (University of Geneva). The GLUTag EC line was a kind gift from Daniel J Drucker (University of Toronto).

## Additional information

### Competing interests

Tommaso Patriarchi: T.P. is a co-inventor on a patent application (WO2018098262A1) related to the sensor technology described in this article. The author has no other competing interests to declare. The other authors declare that no competing interests exist.

### Funding

| Funder | Grant reference number | Author |
|---|---|---|
| European Research Council | 891959 | Tommaso Patriarchi |
| Schweizerischer Nationalfonds zur Förderung der Wissenschaftlichen Forschung | 310030_196455 | Tommaso Patriarchi |
| Schweizerischer Nationalfonds zur Förderung der Wissenschaftlichen Forschung | 310030L_212508 | Tommaso Patriarchi |
| Schweizerischer Nationalfonds zur Förderung der Wissenschaftlichen Forschung | 200021_200865 | Nina Hartrampf |

The funders had no role in study design, data collection and interpretation, or the decision to submit the work for publication.

### Author contributions

Loïc Duffet, Resources, Data curation, Formal analysis, Investigation, Methodology, Writing – original draft, Writing – review and editing; Elyse T Williams, Resources, Data curation, Formal analysis, Investigation, Methodology, Writing – review and editing; Andrea Gresch, Investigation, Writing – review

and editing; Simin Chen, Musadiq A Bhat, Resources, Methodology, Writing – review and editing; Dietmar Benke, Resources, Writing – review and editing; Nina Hartrampf, Conceptualization, Data curation, Formal analysis, Supervision, Funding acquisition, Methodology, Writing – original draft, Project administration, Writing – review and editing; Tommaso Patriarchi, Conceptualization, Data curation, Formal analysis, Supervision, Funding acquisition, Visualization, Methodology, Writing – original draft, Project administration, Writing – review and editing

## Author ORCIDs
Loïc Duffet ⬤ http://orcid.org/0000-0001-6642-3039
Musadiq A Bhat ⬤ http://orcid.org/0000-0002-0894-9996
Nina Hartrampf ⬤ http://orcid.org/0000-0003-0875-6390
Tommaso Patriarchi ⬤ http://orcid.org/0000-0001-9351-3734

## Ethics
Animal procedures were performed in accordance with the guidelines of the European Community Council Directive or the Animal Welfare Ordinance (TSchV 455.1) of the Swiss Federal Food Safety and Veterinary Office and were approved by the Zürich Cantonal Veterinary Office (licence number: ZH087/2022). Rat embryos (E17) obtained from timed-pregnant Wistar rats (Envigo) were used for preparing primary cortical neuronal cultures.

Public Review https://doi.org/10.7554/eLife.86628.3.sa1
Author Response https://doi.org/10.7554/eLife.86628.3.sa2

---

# Additional files

## Supplementary files
• Source data 1. Synthesis and Characterization of Building Blocks and Peptides.
• MDAR checklist

## Data availability
DNA plasmids used for viral production have been deposited both on the UZH Viral Vector Facility (https://vvf.ethz.ch/) and on AddGene (plasmid numbers: 187466-187468). Plasmids and viral vectors can be obtained either from the Patriarchi laboratory, the UZH Viral Vector Facility, or AddGene. Source data are provided with the manuscript.

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
