## [Editor Report · eLife assessment]

This **valuable** Tools and Resources paper presents new tools for
investigating GLP-1 signaling: a genetically-encoded sensor constructed from a
mutated GLP1R receptor as well as a caged agonist peptide. The evidence for these
tools working as advertised is **solid** and they may be useful for
screening compounds that bind to GLP1R.

---

## [Referee Report · Public Review]

This paper presents two new tools for investigating GLP-1 signaling. The genetically
encoded sensor GLPLight1 follows the plan for other GPCR-based fluorescent sensors,
inserting a circularly permuted GFP into an intracellular loop of the GPCR. The
light-uncaged agonist peptide, photo-GLP1, has no detectable agonist activity (as
judged by the GLPLight1 sensor) until it is activated by light. However, based on
the current characterization, it is unclear how useful either of these tools will be
for investigating native GLP-1 signaling.

The GLPLight1 sensor has a strong fluorescent response to GLP-1 with an EC50 of ~10
nM, and its specificity is high, as shown by lack of response to ligands of related
class B GPCRs. However, the native GLP1R enables biological responses to
concentrations that are ~1000-fold lower than this (as shown, for instance, in a
supplemental figure of this paper). This makes it difficult to see how the sensor
will be useful for in vivo detection of GLP-1 release, as claimed; although there
may be biological situations where the concentration is adequate to stimulate the
sensor, this is not established. Data using a GLP-1 secreting cell line suggest that
the sensor has bound some of the released GLP-1, but it is difficult to have
confidence without seeing an actual fluorescence response to stimulated release.

Alternatively, the sensor might be used for drug screening, but it is unclear that
this would be an improvement over existing high-throughput methods using the cAMP
response to GLP1R activation (since those are much more sensitive and also allow
detection of signaling through different downstream pathways).

The utility of the caged agonist PhotoGLP1 is similarly unclear. The data demonstrate
a substantial antagonism of GLP-1 binding by the still-caged compound, and it is
therefore unclear whether the kinetics of the response to PhotoGLP1 itself would
mimic the normal activation by GLP-1 in the absence of the caged compound. A further
concern is that the light-dependence of the agonist effect of PhotoGLP1 was
evaluated only with the GLPLight1 sensor and not with GLP1R signaling itself, which
is 1000x more sensitive and which would be the presumed target of the tool. In
addition, PhotoGLP1 is based upon native GLP-1, which is rapidly truncated and
inactivated by the peptidase DPPIV, expressed in most cell types, and expressed at
very high levels in the plasma. The utility of PhotoGLP1 is therefore limited to
acute (minutes) in vitro experiments.

---

## [Author Response]

The following is the authors' response to the original reviews.

We would like to thank all Reviewers for their careful evaluation of our work. Below
please find our responses and comments.

**Reviewer #1 (Recommendations For The Authors):**
1. The detection of cell-released GLP-1 is addressed in an indirect, averaged way
in Fig. 2 - Supplement 1. This question seems like a good opportunity for an
antagonist experiment (Exendin-9), which presumably would require much lower
concentrations than those used to antagonize a saturating dose of GLP-1. It
would also be much more convincing if GLPLight1 could be used to detect
stimulated release of GLP-1 from the GLUTag cells.

We tried multiple times to acutely stimulate GLUTag cells using Forskolin and IBMX,
but unfortunately we did not observe any robust fluorescence increase of GLPLight1.
The only observation that was consistent was the higher baseline fluorescence of
GLPLight1, and the reduced maximal response to saturating GLP-1 when GLPLight1
expressing HEK cells were cultured overnight with GLUTag cells. We considered this
assay to be at best qualitative and — despite the aforementioned attempts — could
not determine quantitative values.

2. The excitation-ratiometric response of the sensor, shown in Fig. 1D, is
usually accompanied by strong pH-dependence of sensor function. It would be
valuable to characterize this pH-dependence, using permeabilized cells in which
the pH is changed; the ability of small (0.2-0.5 unit) pH changes to produce
changes in fluorescence, as well as to affect the dynamic range of the sensor,
should be characterized. This will prevent the misidentification of agents that
affect cellular pH as having (for instance) an inhibitory effect on the binding
of GLP-1 to GLPLight.

The pH sensitivity of cpGFP-based sensors is a valid concern. However, considering
that the cpGFP module from GLPLight1 is intracellular (and thus largely protected
from potential extracellular pH changes) we assume that GLPLight1 signal should be
robust in most in-vivo or cell-based assays. In fact we have previously
characterized this for a similarly-built neuropeptide sensor (PMID: 35145320) and
believe that this will be the case also for GLPLight1.

3. The reported Kd for Exendin-9 is in the low nM range. Please explain the
partial response at 1000x the concentration (including a discussion of the Kd of
GLP-1 itself, as well as its off kinetics, and a comparison of this assay to the
assays used previously).

The partial response is due to the presence of 1 uM GLP-1 in the imaging buffer,
which is in constant competition with Exendin-9 for the binding to GLPLight1.
Because GLP-1 has similar affinity as Exendin9 (see for example PMIDs: 34351033 and
21210113) and both are present at saturating concentration, we did expect to observe
a partial response from GLPLight1. In this study, we did not exactly determine the
on and off kinetics of both GLP-1 and Exendin9 on the GLPLight1 sensor due to
technical challenges: to perform these experiments, we would need to set up a
perfusion system where we could remove the unbound ligand and either wash off the
bound ligand with buffer or compete it out with an antagonist. Unfortunately, we
currently do not have access to such a set up.

4. Are the turn-on kinetics in Fig. 2C limited by drug application or by
association? Are the on-rates much slower for the lower concentrations used for
Fig. 2C? This is important for knowing how fast responses are likely to be at
the lower concentrations likely to be achieved by endogenous release.

If we consider Fig 2B and 2C, we assumed the on-kinetics to be mostly driven by
association since the ligand is expected to be homogeneously distributed.

The on-rate kinetics are indeed slower when lower concentrations of GLP-1 are used as
shown in (Figure 2b) where we observe a TauOn of 4.7s with 10 uM GLP-1 and much
slower kinetics when GLP-1 is applied a 1 uM for example (Figure 3d). As a result,
we chose to incubate the ligand with GLPLight1 expressing cells for at least 30
minutes before the measurement of the dose-response to be close to equilibrium.

5. The parameters for the fitted dose-response curves in Fig 2C should be listed.
The ~4x discrepancy between the dose-response in HEK-293 cells and neurons
should be discussed. Are there known auxiliary subunits, dimerization, or lipid
dependence that might account for this? It seems important to understand this if
the sensors are to be used in an assay that may compare different systems.

We added the EC50 values to Fig 2C as requested. We did not consider a 4x discrepancy
to be significant, because the measurement error in the EC50 region is relatively
high and this difference seemed to be within the error range. In fact, the 95%
confidence interval ranges are 7.8 to 11.1 nM in Neurons and 23.8 to 32.1 nM for HEK
cells, if we consider the upper and lower boundaries of each, the difference drops
to around 1-fold. We also performed a statistical test to compare the two fits
(Extra sum of squares F-test) that confirmed the two fits were not significantly
different (P value = 0.3736). Of course, the interaction partners and membrane
composition are different in HEK cells and neurons and probably have an influence on
the EC50 of GLPLight1, but their exact influence is unclear.

6. It seems surprising that removal of the endogenous N-terminal secretory
sequence is actually helpful for membrane expression. Do the authors have any
suggested explanation for this?

GLPLight1 contains an N-terminal hemagglutinin (HA) secretory motif. The hmGLP1R
sequence that we chose also contained an endogenous secretory sequence that most
likely interfered with the membrane transport mechanism and resulted in a lower
sensor expression with both secretory sequences. We thus decided to keep the HA
instead of endogenous to remain consistent with other sensors created in-house.

7. In Fig. 1, supplement 3, are the transient responses real? Do they occur with
the control construct?

While we have not measured the G-protein recruitment on GLPLight-ctr, we have often
observed this phenomenon for various receptors and ligands. The transient responses
are thus most likely an artifact after manual addition of the ligand possibly due
to:

Temperature differenceExposure of the plate to ambient light before resuming measurement
(phosphorescence)Re-suspension of the cells affecting the proximity to the detectorOther unknown variables

If these responses were real, we would also expect them to be more sustained over
time.

8. Please include a sentence or two explaining the luminescence complementation
assay, and a reference.

We updated the results section of the manuscript with a section describing the
luminescence complementation assay along with a reference:

*“Next, we compared the coupling of GLPLight1 and its parent receptor (WT
GLP1R) to downstream signaling. We first measured the agonist-induced membrane
recruitment of cytosolic mini-G proteins and β-arrestin-2 using a split
nanoluciferase complementation assay (Dixon et al., 2016). In this assay both
the sensor/receptor and the mini-G proteins contains part of a functional
luciferase (smBit on the sensor/receptor and LgBit for Mini-G proteins) that
becomes active only when these two partners are in close proximity (Wan et al.,
2018).”*

Bravo to the authors for already making the sensor plasmids available at
addgene.com. It would be helpful to include the plasmid IDs and/or a URL in the
manuscript.

We would like to thank Reviewer #1 for noticing this. We have updated the data
availability section of the manuscript and added the AddGene plasmid numbers of the
constructs generated in this study.

**Reviewer #2 (Recommendations For The Authors):**
1. There are some parts of the introduction that need clarification. For example,
GLP1 is quoted as an anorexigenic peptide, however, that is probably only true
for centrally- derived GLP1. There is no evidence that enteroendocrine-derived
GLP1 (the major pool) is anorexigenic- it is likely to be substantially degraded
by DPPIV before reaching the brain. In any case, the discovery of GLP1 was
always one of glucose-dependent insulin secretion, with the brain system being
described decades later. Overall, the intro needs to be slightly reframed. While
the tools presented here are more useful for assessment of central
GLP1-releasing circuitry, they are ultimately based upon GLP1R signaling that is
much better validated in the periphery.

We have slightly reframed the introduction accordingly.

2. "The human GLP1R (hmGLP1R) is a prime target for drug screening and drug
development efforts, since GLP-1 receptor agonists (GLP1RAs) are among the most
effective and widely-used weight-loss drugs available to date (Shah and Vella,
2014)." GLP1R was for two decades the breakthrough drug for treatment of
type 2 diabetes mellitus and correction of glucose tolerance as assessed through
HbA1c. It is only through reporting on millions of patients receiving GLP1RA
that the weight loss effects were noted, leading to Phase1-3 trials and eventual
approval for obesity indication. Again, some slight reframing of the
introduction is required here.

Also for this point, we have slightly reframed the introduction accordingly.

3. GLP1 was applied at a maximal dose of 10 uM, which is 10-fold higher than
maximal. Can the authors confirm absence of cytotoxic effects of exposing to
peptide at such concentration? Ex4 (9-39) at such concentrations is usually
cytotoxic at least in primary tissue.

We did not observe any obvious cytotoxic effect of GLP-1 at this concentration in
HEK293T cells or Neurons.

4. "As expected, GLPLight1 responded to both GLP1RAs with almost maximal
activation, on par with GLP1 (Figure 2a)." Such a claim is difficult to
interpret without concentration-response curves, since the maximal concentration
of liraglutide and semaglutide might not have been achieved in these
experiments.

We agree with this statement is difficult to interpret without further clarification.
We know from the literature that GLP-1, liraglutide and semaglutide all have very
high affinity to the hmGLP1R (PMID: 31031702). We also proved that GLPLight signal
saturates at concentrations above 1 uM of GLP-1 (figure 2C), we thus applied a 10x
excess of all ligands and considered this signal as maximal.

5. "These results indicate that GLPLight1 can serve as a direct readout of
pharmacological drug action on the hmGLP1R with higher temporal resolution than
previously available approaches, such as downstream signaling assays (Zhang et
al., 2020)." Many investigators use cAMP imaging to investigate GLP1R
signaling, which is arguably of similar spatiotemporal resolution, also with the
advantage of FRET quantification in some cases (e.g. EpacVV). Direct GLP1R
signaling can also be inferred using cell lines heterologously-expressing GLP1R.
Thus, the advantage of the current probes is that they can be used to readout
direct GLP1R activation in native cells/tissues where promiscuous class B
binding might limit signaling measures or where endogenous GLP1 release needs to
be investigated.

We have edited the manuscript text accordingly.

6. "State-of-the-art techniques for detecting endogenous GLP-1 or glucagon
release in vitro from cultured cells or tissues consist of costly and
time-consuming antibody- based assays (Kuhre et al., 2016) or analytical
chemistry procedures (Amao et al., 2015)." Agreed, but
non-specificity/cross-reactivity of such assays is more prohibitive/problematic
(e.g. against glicentin).

We have edited the introduction accordingly.

7. The studies using co-culture of GLUTag and GLP1Light1-HEK293 cells, whilst
interesting, are not entirely convincing in their current form. Firstly,
co-culture could influence GLP1Light expression levels (can the authors label
FLAG?). Secondly, specificity of the response is not tested e.g. by adding Ex4
(9-39). Thirdly, titration with GLUTag conditioned media is not performed.

We partially addressed this issue in the answer to comment #1 from Reviewer #1. We
previously performed a FLAG staining of GLPLight1 in the presence or absence of
GLUTag cells and we did not notice any obvious difference. This goes in line with
the fact that GLPLight1 is signaling inert, and the presence of GLP1 should not
interfere with the surface expression of the sensor. We also checked that HEK293T
cells did not express high levels of GLP1R according to the BioGPSCell line Gene Expression profile.

We also tried to add GLUTag media after stimulation in bolus to GLPLight1 expressing
cells and observed no response. This indicated that the “sniffer” cells must be
present in close proximity to GLUTag cells for an extended period of time to observe
any substantial difference in response, justifying our choice of experimental
setup.

8. "Given that our photocage was placed at the very N-terminus of
photo-GLP1, our results show that this caging approach prevents the peptide's
ability to activate GLP1R but, at the same time, preserves its ability to
interact with the ECD." An alternative hypothesis is that PhotoGLP1 does
activate GLP1R, but this is undetectable with the sensitivity of GLP1Light.
PhotoGLP1 cAMP concentration-response assays are needed (uncaged versus cage) to
properly characterize and validate the compound (as would be standard for any
newly-described GLP1R peptide ligand).

While we agree that there is a chance that Photo-GLP1 could activate GLP1R at high
concentrations, we think that the characterization of Photo-GLP1 has to be
determined by the end user directly with the technique of choice (GLPLight1 in our
case) in order to get a reliable comparison of potency and efficacy. We modified the
text accordingly to more accurately reflect the direct conclusions from our data, as
follows:

*“our results show that this caging approach prevents the peptide's ability to
activate GLPLight1”.*

9. "Surprisingly, GLPLight1 shows a fluorescent response in all three
uncaged areas, while its fluorescence remained unaltered throughout the rest of
the FOV, indicating high spatial localization of the response to GLP-1 (Figure
3f)." Why is this surprising?

We agree that this result is, indeed, not surprising and would like to thank Reviewer
#2 for spotting this mistake, which has now been corrected in the manuscript.

10. The localized PhotoGLP1 experiments are interesting and show the utility of
the ligand. There is however activation outside of the region of uncaging, which
would argue against a pre-bound ECD mode of action. Possibly some PhotoGLP1 is
pre- bound to the ECD, and some is freely diffusing? Alternatively, the scan
area might be below the diffraction limit/accuracy of the microscope?

We would like to thank Reviewer #2 for this comment and agree with their observation.
There could be some free Photo-GLP1 that gets photo-activated and binds regions
around the uncaging area (similar to what has been observed for Photo-OXB:,PMID:
36481097). The activation around the uncaging area could also be due to lateral
diffusion of the activated receptor on the membrane. There is also most likely some
light diffraction at the uncaging area that could account for this phenomenon. To
increase the spatial resolution, future studies could involve uncaging during sensor
imaging via two-photon microscopy.

11. What was the rationale for caging native GLP1, which is then susceptible to
DPPIV-mediated degradation? Would the N-terminal cage and first 2 amino acids
also not be cleaved by DPPIV, thus rendering the tool of limited in vivo
application? Conversely, PhotoGLP1 provides a template for similar
light-activated (stabilized) GLP1R agonists such as Ex4 or liraglutide.

Thank you for making us aware of this (in vivo) limitation. We designed photoGLP1 as
a tool for neurobiological experiments in the brain, where DPPIV expression would be
low compared to peripheral organs (https://www.proteinatlas.org/ENSG00000197635-DPP4/tissue). We also
envisage that the presence of the photocage would be enough to hinder the binding to
DPP4 that cuts the first 2 AA. This hypothesis, however, was never tested
experimentally, and we, therefore, acknowledge the limitation in the manuscript. We
would furthermore like to thank the reviewers for his comment on additional
photo-caged GLP1 agonists, which could be developed future studies.

12. It wasn't clear how GLP1Light could be used as a HTS screen for drug
discovery? Surely, conventional systems (e.g. GLP1R + BAR/Ca2+/cAMP reporting)
allow signal bias, an important component of GLP1RA action, to be assessed. Or
could GLP1Light1 be used as a pre-screen to exclude any ligands that do not
orthosterically bind GLP1R?

We would like to thank Reviewer #2 for this comment and would like to offer some
clarification. We indeed thought that GLPLight1 could be used as a first line of
screening to exclude ligands that do not bind in the orthosteric pocket. It is also
a rather flexible method as the fluorescence increase of those sensors can be
monitored using various techniques/devices that are available in most labs (e.g.
microscopy, plate reader, flow cytometry).

13. Limitations of GLP1Light1 and PhotoGLP1 are not acknowledged in the
discussion.

We would like to thank Reviewer #2 for pointing out the lack of description of the
limitations of these tools, which have now been added to the Discussion.

14. Full characterization of PhotoGLP1 is missing, to include UV/Vis, Tr and
HRMS.

PhotoGLP1 was fully characterized by UV/Vis and HRMS, and all experimental and
analytical data was uploaded as supplementary data when the manuscript was initially
submitted for publication in eLife.

**Reviewer #3 (Recommendations For The Authors):**
1. The ~1000 fold lower EC50 for GLP1 of GLPLight1 compared with native GLP1R
needs to be openly acknowledged as a major limitation of the sensor, as this
will substantially reduce the types of experiment for which it will be useful.
Because it needs 1000 times higher GLP1 levels than wild type GLP1R to be
activated, it is unlikely, for example, to be useful for monitoring the dynamics
of activation of native GLP1R in vivo. The claim that the sensor could be used
for in vivo imaging for fibre photometry is therefore an exaggeration.

We would like to first thank Reviewer #3 for this comment and to further provide some
clarification. We recognized that the data presented in this manuscript might have
been confusing when comparing the affinity of GLP1R (using cAMP) and GLPLight1
(using the fluorescence increase because there is no coupling to cAMP). We believe
that the low EC50 measured in the cAMP assay cannot accurately be compared to
GLPLight1 response because it is an enzymatically amplified process. In order to
support this claim, we included another set of experiments where we titrated
agonist- induced recruitment of miniGs protein to the GLP1R receptor and found an
EC50 of 3.8 nM for native GLP-1 using this assay (added as panel l in Figure1
Supplement 3). We thus confirmed that the nature of the assay itself has a drastic
influence on the EC50 measured and it is not unusual to observe 100x fold difference
of EC50 for the same receptor-ligand pair.

We believe that the miniGs protein recruitment is a better comparison to GLPLight1
because it is not enzymatically amplified. This assay reveals that GLPLight1 has
around 8-fold lower affinity to GLP1 compared to its parent receptor, which is in
line with the EC50 loss observed previously for other GPCR-based sensors of this
class. We are thus confident that GLPLight1 has to potential to be used in vivo
under specific circumstances, specifically in brain tissue. We elaborated on this
point in the Discussion part of the manuscript.

2. Fig2 suppl 1 is described as demonstrating a reduced response of GLPLight1 to
GLP-1 when HEK cells with were cultured with GLUTag cells. However, it is
speculation to conclude that this is because GLP1Light1 was partially
pre-activated by endogenous GLP-1, without demonstrating the response of
GLPLight1 before and after GLUTag cell stimulation. Unless additional data are
generated, the presented data do not convincingly demonstrate that GLP1Light1
can detect GLP1 released from GLUTag cells.

We would like to thank Reviewer #3 for this comment which has been addressed already
in the replies to Comment#1 from Reviewer #1 and Reviewer #2.

3. The authors should openly acknowledge that photo-uncaging the GLP1 probe might
not be very helpful for monitoring the temporal dynamics of the GLP1-GLP1R
interaction, because unless all the photocaged glp1 is released by the light
stimulus, the activation of photo-released GLP1 will be slowed by the remaining
caged GLP1, and the dynamics will be slower than for native GLP1. This makes it
unsuitable for many temporal questions, although it might be useful to deliver
GLP1 in a spatial restricted manner.

We do agree that the biggest advantage of Photo-GLP1 is its ability to be activated
in a very localized manner. We also agree that the presence of caged Photo-GLP1 will
influence the binding of the uncaged GLP-1. Nevertheless, there is still an
advantage of using Photo-GLP1 in some assays such as pharmacological activation on
brain slices. In fact, we have shown for our Photo-OXB molecule that the perfusion
of OXB was much slower at eliciting neuronal depolarization compared to uncaging of
Photo- OXB (see PMID: 36481097). We think that this was mainly due to the slow
diffusion kinetics of the peptide into the brain tissue. We also think that uncaging
can provide a more controlled activation with varying laser power and uncaging
duration.

4. To claim (as currently in the discussion) that GLPLight1 has potential to be
used for investigating the dynamics of endogenous GLP1, the authors would need
to compare the dynamics of the GLP1Light sensor with wild type GLP1R. We do not
know that its activation dynamics will reproduce native glp1r.

We would like to thank Reviewer #3 for this comment and would like to offer some
clarification. Since GLPLight1 does not couple to intracellular signaling, it was
impossible to compare its activation kinetics to GLP1R WT using the same assay.
However, we can offer a relative comparison since we know that GLPLight1 takes
around 50 seconds to be activated using 1 µM GLP-1 (figure 2B) and that it takes a
similar time for GLP1R to be activated in the miniG protein recruitment assay (Fig 1
Supplement 3) using 100 nM GLP-1. Considering that GLPLight1 has a lower affinity
than the GLP1R (8-10x lower), we think that the activation kinetics of both the
sensor and GLP1R are comparable.

Additional comments:1. In fig 2A,B, it is not clear whether the trace shows a partial reversal of
GLP1- triggered activation by Ex9, or Ex9-independent receptor desensitization.
A control trace is required to show the kinetics of GLP1-triggered activation
without the addition of Ex9.

We would like to thank Reviewer #3 for this comment. We can exclude the possibility
of Ex9-independent desensitization because GLPLight1 has been shown to be signaling
inert to all G-proteins, Beta arrestin-2 and cAMP. Moreover, we have observed that
the fluorescence signal was stable for more than 30 minutes for the GLP-1
titrations, even at high concentrations of ligand.

2. It would be helpful if the pEC50 for WT GLP1 were also shown in table 1, for
comparison with the GLP1 mutants.

We would like to thank Reviewer #3 for this comment, and we have now added the
respective pEC50 for WT GLP1 to Table 1.

3. Fig2 suppl 1. The methods and analysis for this figure are inadequately
explained. To show that the HEK-GLPLight1 cells are responding to GLP1 released
from GLUTag cells, the GLPLight1 response needs to be shown before and after
GLUTag cell stimulation with an agent that should trigger GLP-1 release.

We would like to thank Reviewer #3 for this comment which has been partially
addressed already in the replies to Comment#1 from Reviewer #1 and Reviewer #2.

Since we did not observe any response to acute stimulation of GLUTag cells we
considered the high glucose concentration present in the culture media being a
stimulation agent for GLUTag cells, which has been previously reported (PMID:
17643200).

4. Fig 3g and others: The end of the photo activation period needs to be
represented correctly on the timeline. In 3g, the bar that should indicate when
photoactivation was applied does not end at the zero time point (which is
labelled as the time relative to photoactivation).

We would like to thank Reviewer #3 for pointing this out. The shaded area
representing the photo-activation has been matched accordingly.

5. Discussion para 1: the authors claim their data show that ligand induced
activation of human GLP1R occurs more slowly than others similar GPCR sensors -
they should give actual data to substantiate this claim, since the time course
of glp1r activation has not been analysed and compared with other sensors in the
manuscript.

We added data to support this claim to the discussion: *“As a reference, other
previously-characterized class-A GPCR-based neuropeptide biosensors showed sub-
second activation kinetics (Duffet et al., 2022a; Ino et al.,
2022).”*

6. Methods: what wavelength was used for recording emission from GLP1Light1? The
excitation wavelength is given, but I can't see the emission wavelength(s). In
fig 1d, the excitation and emission spectra should be depicted in different
colours/line properties, otherwise this figure is very confusing.

We updated figure1d and changed the colors to improve data visualization. Regarding
the missing wavelength, we would like to clarify that both wavelengths were already
described in the methods section as: *“The excitation and emission spectra
were measured at λem = 560nm and λex = 470nm, respectively, on a TECAN M200 Pro
plate reader at 37 °C. “.* We would be happy to rewrite this paragraph,
if necessary, shall it remain unclear to the reader.